# Associations of social environment, socioeconomic position and social mobility with immune response in young adults: the Jerusalem Perinatal Family Follow-Up Study

Gabriella M Lawrence,[1] Yehiel Friedlander,[1] Ronit Calderon-Margalit,[1] Daniel A Enquobahrie,[2,3] Jonathan Yinhao Huang,[3,4] Russell P Tracy,[5] Orly Manor,[1] David S Siscovick,[6] Hagit Hochner[1]

For numbered affiliations see end of article.

**Correspondence to**
Dr Hagit Hochner;
hagith@ekmd.huji.ac.il

## ABSTRACT

**Objectives** Immune response to cytomegalovirus (CMV) impacts adult chronic disease. This study investigates associations of childhood and adulthood social environment, socioeconomic position (SEP) and social mobility with CMV response in young adults.

**Design** Historical prospective study design.

**Setting** Subcohort of all 17 003 births to residents of Jerusalem between 1974 and 1976.

**Participants** Participants included 1319 young adults born in Jerusalem with extensive archival and follow-up data, including childhood and adulthood SEP-related factors and anti-CMV IgG titre levels and seroprevalence measured at age 32.

**Main exposure and outcome measures** Principal component analysis was used to transform correlated social environment and SEP-related variables at two time points (childhood and adulthood) into two major scores reflecting household (eg, number of siblings/children, religiosity) and socioeconomic (eg, occupation, education) components. Based on these components, social mobility variables were created. Linear and Poisson regression models were used to investigate associations of components and mobility with anti-CMV IgG titre level and seroprevalence, adjusted for confounders.

**Results** Lower levels of household and socioeconomic components in either childhood or adulthood were associated with higher anti-CMV IgG titre level and seropositivity at age 32. Compared with individuals with stable favourable components, anti-CMV IgG titre level and risk for seropositivity were higher in stable unfavourable household and socioeconomic components (household: β=3.23, P<0.001; relative risk (RR)=1.21, P<0.001; socioeconomic: β=2.20, P=0.001; RR=1.14, P=0.01), downward household mobility (β=4.32, P<0.001; RR=1.26, P<0.001) and upward socioeconomic mobility (β=1.37, P=0.04; RR=1.19, P<0.001). Among seropositive individuals, associations between household components and mobility with anti-CMV IgG titre level were maintained and associations between socioeconomic components and mobility with anti-CMV IgG titre level were attenuated.

### Strengths and limitations of this study

► Cytomegalovirus (CMV) antibody as a biomarker for atherosclerosis is a novel addition to studies of cardiometabolic risk.

► This is a unique population-based birth cohort with rich archival and follow-up data from birth to young adulthood.

► We used an aggregate view of the social environment and socioeconomic position based on high-quality characterisation in both childhood and adulthood to assess relationships at different time points in life as well as changes from childhood to adulthood with adult immune response reflected by CMV titres.

► Timing at initial seroconversion cannot be determined as only a single CMV titre measure was taken at young adulthood, also limiting precision in detection of lifetime viral load.

**Conclusions** Our study provides evidence that accumulating low SEP from childhood through adulthood and social mobility may compromise immune response in young adulthood.

## INTRODUCTION

Chronic inflammation is a known risk factor for cardiovascular disease, and infection-induced inflammation may be one of the causes of endothelial dysfunction leading to atherosclerosis.[1] Specifically, immune response to cytomegalovirus (CMV), a highly transmissible beta herpesvirus endemic throughout the world[2] and a biomarker of immune response,[3 4] has been implicated in the development of atherosclerosis[3 5–7] and cardiovascular-related morbidity[8 9] and mortality.[5 10–12] Until recently, non-congenital CMV was not seen as clinically relevant in immunocompetent individuals due to the lack of overt

clinical symptoms. However, accumulating evidence suggests that CMV antibody titre is associated with proinflammatory cytokines[3 4]; more memory T cells are devoted to suppression of latent CMV[12 13] and fewer naïve T cells are available to respond to new infections,[12–14] leading to chronic immune activation and inflammation, and subsequent cardiovascular disease. Despite the sizeable cellular response from the host, a characteristic unique to CMV, the virus adapts to the immune system efficiently. CMV, therefore, is never eliminated from the infected immunocompetent individual, in whom it causes a persistent asymptomatic infection.[15 16]

Research suggests that early-life environments affect immune function over the life course.[17] Studies show that early-life stress results in infection-related premature death, immunosuppression,[18] impairment in components of cell-mediated immunity, reduction in antibody response to vaccination and elevated antibody level[17 19 20] during childhood and beyond. A life course approach to health proposes that interaction of environment and biological factors (eg, epigenetic programming) experienced throughout the life course impact current and future health.[21–23]

Additionally, research has clearly indicated that socioeconomic factors throughout the life course impact adult health[21 24–30] and particularly in Israel, religious observance, a central characteristic of the Israeli social environment, contributes independently to social position and adult health.[31] Associations between CMV seropositivity and antibody titre with education, income,[32] household crowding[2 33] and family size[34] further indicate that social factors influence immune response. Additionally, recent studies examining the impact of social mobility and socioeconomic gradients on adult health report varied results.[24 25 35 36] While upward mobility may attenuate the harmful effects of earlier disadvantage, and downward mobility is hypothesised to have a detrimental influence on health despite advantages at earlier ages, sustained exposure to low socioeconomic position (SEP) is most commonly associated with worse health.[25 36 37] Though associations between early-life socioeconomic factors and CMV have been demonstrated, the effect of social transitions from childhood to adulthood on CMV response has not been investigated.

This study investigates associations between SEP in childhood and adulthood and social mobility from birth to 32 years with CMV response in young adults of the Jerusalem Perinatal Family Follow-Up Study.

## METHODS

The Jerusalem Perinatal Study (JPS) population-based cohort includes a subcohort of all 17 003 births to residents of Jerusalem between 1974 and 1976.[38 39] Data consist of demographic and socioeconomic information, maternal medical conditions during current and previous pregnancies, and offspring birth weight, abstracted from birth certificates or maternity ward logbooks. Additional information, including socioeconomic-related factors, gestational age, smoking, pre-pregnancy and end of pregnancy weight, and gynaecological history, was collected by interviewing mothers on first or second postpartum day. Detailed information on data collection is previously described.[38 39]

The JPS Family Follow-Up Study includes a sample of 1400 offspring from the 1974–1976 cohort interviewed and examined between 2007 and 2009 (mean age=32; range: 30–35 years).[40] Briefly, sampling frame included singletons and term (gestational age ≥36 weeks) births without congenital malformations stratified by maternal pre-pregnancy body mass index and birth weight. Blood samples at fasting were taken and offspring IgG antibodies to CMV antigen were measured in plasma using indirect enzyme immunoassay with the Diamedix Immunosimplicity Is-CMV IgG Test Kit (Diamedix, 2008) at the University of Vermont. Further test kit details are provided in online supplementary 1. Anti-CMV IgG titre levels were measured for 1319 offspring with complete data.

All participants provided informed consent.

### Study variables
#### Outcome variables
The primary outcome examined was offspring CMV response measured through offspring antibody to CMV at age 32. Transformation (square root) of anti-CMV IgG titre level (EU/mL) was performed due to right skewness and treated as a continuous variable. CMV seroprevalence (seropositivity ≥10 EU/mL) was examined as a secondary dichotomous outcome.

#### Explanatory variables
The multiethnic society of Israel[41] provides an opportunity to examine SEP alongside culturally relevant social environment characteristics, including religiosity and family size in addition to well-recognised SEP variables such as occupation and education.

SEP-related explanatory variables were examined at birth and age 32, reflecting early (ie, prenatal and perinatal periods) and young adult life course periods, respectively. Variables at childhood include: (1) maternal religiosity (self-identified as ultraorthodox, orthodox, traditional, secular and collapsed into dichotomous categories: ultraorthodox and orthodox as religious, traditional and secular as not religious); (2) paternal lay leadership (ie, whether father was an ordained rabbi, dichotomous); (3) father's occupation (scale: 1–6; 1—low, 6—high) as described by Corcoran et al[26]; (4) maternal education (years, continuous); and (5) number of siblings (continuous). Variables at adulthood include: (1) offspring religiosity (self-identified as ultraorthodox, orthodox, traditional, secular and collapsed into dichotomous categories: ultraorthodox and orthodox as religious, traditional and secular as not religious); (2) offspring occupation (scale: 1–6; 1—low, 6—high); (3) offspring education (years, continuous); and (4) parity at age 32 (continuous).

Potential confounders include offspring sex (dichotomous), maternal and paternal age at childbirth (years, continuous), maternal and paternal smoking during pregnancy and offspring smoking in young adulthood (number of cigarettes smoked per day: 0, 1–10, 11–20, 21+).

### Principal component analysis

Following the successful use of principal component analysis (PCA) in creating socioeconomic indices in past and current studies,[42 43] and given the available array of SEP-related variables, including SEP variables well known in the literature as well as specific social environment-related variables mentioned above, and the overlap and correlation between SEP-related variables (including individual-level indicators such as education, occupation-based indicators and life course SEP),[44] we used PCA to reduce the number of potentially correlated variables in our cohort to a smaller number of principal components. (See online supplementary 2 for correlations between SEP-related variables in the JPS Family Follow-Up cohort.) PCA[45] carried out separately for variables measured at childhood and adulthood, transformed SEP-related variables into two major components: the social environment reflected by household-related characteristics (eg, number of siblings/children, religiosity) and the socioeconomic (eg, occupation, education) environment. The two components together explained 74% of childhood and 75% of adulthood variance. In addition to capturing the largest possible variance, these components nicely portray two elements of SEP in Israeli society—the socioeconomic environment represented by parental and offspring education and occupation as is often the case in many societies, and the social environment, unique to Israeli society, in which families that identify as more religious tend to have more children,[46] with the most religious families often among those with lower income level.[47] Detailed information on component construction and variables included in PCA are provided in online supplementary 3. Higher component score indicates improvement in SEP—that is, higher household component score indicates fewer children/decreased religiosity and higher socioeconomic component score indicates higher occupation level/years of education.

### Social mobility

For both household and socioeconomic components, childhood to adulthood social mobility variables were created. Components were divided into tertiles and dichotomous variables were created for each component (lowest tertile=low; middle and high tertiles=high). Four categories of childhood to adulthood transition were created separately for household and socioeconomic component mobility: stable favourable (high-high), upward mobility (low-high), downward mobility (high-low) and stable unfavourable (low-low).

### Statistical analyses

Analysis of variance model, adjusted for sex, was used to compare mean anti-CMV IgG titre level by categories of characteristics obtained at birth and age 32. Linear regression models were used to investigate associations of childhood and adulthood household and socioeconomic components as well as social mobility with anti-CMV IgG titre level measured at age 32. Multivariable models were fitted to assess associations between childhood and adulthood household and socioeconomic components, independent of each other, with anti-CMV IgG titre controlling for sex, parental ages at offspring birth, and parental and offspring smoking. Multivariable models were also fitted to examine associations between individual socioeconomic and household-related variables (ie, father's occupation, maternal education, maternal religiosity, paternal lay leadership, number of siblings, offspring occupation, offspring education, offspring religiosity and parity), independent of each other, with anti-CMV IgG titre controlling for sex, parental ages at offspring birth, and parental and offspring smoking. Similar multivariable models were fitted to assess associations between household and socioeconomic mobility with anti-CMV IgG titre controlling for sex, parental ages at offspring birth, and parental and offspring smoking. Poisson regression models with robust error variance were used to estimate relative risk of CMV seropositivity for each unit increase in component scores and among categories of social mobility variables. In order to verify that associations between socioeconomic factors and CMV titre levels are not primarily a reflection of associations between serostatus and CMV, all analyses were carried out for the total population (both seropositive and seronegative) and additionally among only the CMV seropositive population.

All models used inverse probability weighting to account for stratified sampling. Analyses were carried out using Stata V.12.0 (StataCorp, College Station, TX).

### RESULTS

Parental and offspring characteristics obtained at birth and offspring characteristics at age 32, including anti-CMV IgG titre, CMV seroprevalence and social mobility are listed in table 1. Mean anti-CMV IgG titre and seropositivity were greater among women than men. Overall, 79.7% of the study population was seropositive for CMV. For social mobility, 7.2% of the population was categorised as downwardly mobile in household component and 15.2% experienced downward socioeconomic mobility; 7.1% and 16.9% experienced upward household and socioeconomic mobility, respectively.

Table 2 presents mean anti-CMV IgG titre level by categories of characteristics obtained at birth and age 32 as well as social mobility among the total population. Among the total population, mean anti-CMV IgG was highest among offspring born to ultraorthodox mothers and fathers holding religious leadership positions, as well as offspring born to mothers with <9 years of education and

| Table 1 Study characteristics at birth and age 32* (n=1319) | | |
|---|---|---|
| Cytomegalovirus | | |
| Anti-CMV IgG titre level† | | |
| Female | 12.0 | (7.9) |
| Male | 9.5 | (7.7) |
| Total | 10.8 | (7.9) |
| CMV seropositivity (%)† | | |
| Female | 84.4 | |
| Male | 75.0 | |
| Total | 79.7 | |
| Characteristics obtained at birth | | |
| Maternal religiosity (%) | | |
| Ultraorthodox | 18.4 | |
| Orthodox | 22.2 | |
| Traditional | 35.2 | |
| Secular | 24.2 | |
| Paternal lay leadership (%) | | |
| Yes | 14.0 | |
| No | 86.0 | |
| Paternal occupation level (%) | | |
| Low | 40.0 | |
| Middle | 23.3 | |
| High | 36.7 | |
| Maternal education (years) | 11.6 | (3.4) |
| Number of siblings (%) | | |
| ≤2 | 30.0 | |
| 3–4 | 39.5 | |
| 5+ | 30.5 | |
| Maternal age at childbirth (years) | 28.1 | (5.6) |
| Paternal age at childbirth (years) | 32.0 | (6.9) |
| Maternal smoking in pregnancy (%) | | |
| No | 86.7 | |
| Yes | 13.3 | |
| Paternal smoking in pregnancy (%) | | |
| No | 56.6 | |
| Yes | 43.4 | |
| Characteristics obtained at age 32 | | |
| Religiosity (%) | | |
| Ultraorthodox | 20.5 | |
| Orthodox | 20.8 | |
| Traditional | 20.6 | |
| Secular | 38.1 | |
| Occupation level (%) | | |
| Low | 10.8 | |
| Middle | 33.2 | |
| High | 56.0 | |
| Years of education | 15.5 | (3.9) |
| | | Continued |

| Table 1 Continued | | |
|---|---|---|
| Parity (%) | | |
| ≤2 | 62.6 | |
| 3–4 | 22.8 | |
| 5+ | 14.6 | |
| Smoking (%) | | |
| No | 59.4 | |
| Yes | 40.6 | |
| Social mobility‡ | | |
| Household mobility (%)§ | | |
| Stable and favourable | 58.8 | |
| Upward mobility | 7.1 | |
| Downward mobility | 7.2 | |
| Stable and unfavourable | 26.9 | |
| Socioeconomic mobility (%)§ | | |
| Stable and favourable | 49.5 | |
| Upward mobility | 16.9 | |
| Downward mobility | 15.2 | |
| Stable and unfavourable | 18.4 | |

*Values are expressed as mean (SD) or per cent.
†Obtained at age 32; square root transformed (EU/mL); seroprevalence=IgG titre level ≥10 EU/mL.
‡Social mobility variables represent transition from childhood to adulthood using dichotomised household and socioeconomic components.
§Household and socioeconomic components created using principal component analysis detailed in the Methods section; higher component score indicates improvement in household or socioeconomic factors.
CMV, cytomegalovirus.

those with ≥5 siblings. Similarly, ultraorthodox offspring at age 32, those with ≤12 years of education and with more than five children had higher mean anti-CMV IgG titre level. Among CMV seropositive individuals, a similar pattern of mean anti-CMV IgG titre level by categories of characteristics obtained at birth and age 32 was found (online supplementary 4).

In order to model associations and adjust for additional covariates (ie, offspring sex, parental ages at birth, and parental and offspring smoking), we investigated associations between household and socioeconomic components with anti-CMV IgG titre using linear regression models and with CMV seroprevalence using Poisson regression models presented in table 3.

Lower levels of household and socioeconomic components in childhood and adulthood were associated with higher anti-CMV IgG titre level at age 32 (table 3, continuous CMV model 1) and increased CMV seropositivity (table 3, seroprevalence model 1). When childhood and adulthood components were included together in one model (table 3, continuous CMV model 2), only adulthood household and childhood socioeconomic components were associated with anti-CMV IgG titre (β=−1.02,

**Table 2** Mean* anti-CMV IgG titre level† by categories of characteristics obtained at birth and age 32 (n=1319)

| | Mean | SE | P value‡ |
|---|---|---|---|
| **Characteristics obtained at birth** | | | |
| Maternal religiosity | | | <0.0001 |
| Ultraorthodox | 12.9 | 0.52 | |
| Orthodox | 11.4 | 0.47 | |
| Traditional | 10.5 | 0.37 | |
| Secular | 8.5 | 0.43 | |
| Paternal lay leadership | | | <0.0001 |
| Yes | 13.5 | 0.23 | |
| No | 10.3 | 0.58 | |
| Paternal occupation level | | | 0.8 |
| Low | 10.4 | 0.34 | |
| Middle | 10.6 | 0.44 | |
| High | 10.6 | 0.35 | |
| Maternal education (years) | | | 0.003 |
| <9 | 11.9 | 0.47 | |
| 9–12 | 10.8 | 0.33 | |
| >12 | 9.9 | 0.36 | |
| Number of siblings | | | <0.0001 |
| ≤2 | 9.4 | 0.38 | |
| 3–4 | 10.3 | 0.34 | |
| 5+ | 12.8 | 0.40 | |
| **Characteristics obtained at age 32** | | | |
| Religiosity | | | <0.0001 |
| Ultraorthodox | 12.6 | 0.48 | |
| Orthodox | 12.2 | 0.48 | |
| Traditional | 11.0 | 0.48 | |
| Secular | 9.1 | 0.34 | |
| Occupation level | | | 0.01 |
| Low | 11.4 | 0.68 | |
| Middle | 11.6 | 0.39 | |
| High | 10.2 | 0.29 | |
| Years of education | | | 0.001 |
| ≤12 | 11.9 | 0.42 | |
| >12 | 10.3 | 0.25 | |
| Parity | | | <0.0001 |
| ≤2 | 9.6 | 0.27 | |
| 3–4 | 12.5 | 0.45 | |
| 5+ | 13.1 | 0.57 | |

*Sex adjusted.
†Square root transformed (EU/mL).
‡Obtained through analysis of variance (ANOVA).
CMV, cytomegalovirus.

P<0.001 and β=−0.74, P=0.001, respectively). Among seropositive individuals, associations between household components and anti-CMV IgG titre level were

**Table 3** Regression coefficients, relative risks and 95% CI of CMV in relation to household and socioeconomic components* in childhood and adulthood

| | Continuous CMV† | | | | | | Seroprevalence‡ | | | | | |
|---|---|---|---|---|---|---|---|---|---|---|---|---|
| | Model 1§ | | | Model 2§ | | | Model 1§ | | | Model 2§ | | |
| Component* | Coefficient¶ | P | 95% CI | Coefficient¶ | P | 95% CI | RR¶ | P | 95% CI | RR¶ | P | 95% CI |
| **Household** | | | | | | | | | | | | |
| Childhood | −0.78 | <0.001 | −1.11 to 0.45 | −0.07 | 0.76 | −0.53 to 0.39 | 0.96 | 0.001 | 0.94 to 0.98 | 1.00 | 0.86 | 0.97 to 1.04 |
| Adulthood | −1.25 | <0.001 | −1.61 to 0.90 | −1.02 | <0.001 | −1.55 to 0.49 | 0.94 | <0.001 | 0.91 to 0.97 | 0.95 | 0.02 | 0.91 to 0.99 |
| **Socioeconomic** | | | | | | | | | | | | |
| Childhood | −1.04 | <0.001 | −1.43 to 0.65 | −0.74 | 0.001 | −1.17 to 0.31 | 0.91 | <0.001 | 0.88 to 0.95 | 0.94 | 0.001 | 0.90 to 0.98 |
| Adulthood | −0.71 | <0.001 | −1.09 to 0.33 | −0.39 | 0.08 | −0.82 to 0.04 | 0.93 | <0.001 | 0.90 to 0.97 | 0.95 | 0.02 | 0.92 to 0.99 |

*Created using principal component analysis detailed in the Methods section.
†Continuous square root transformed (EU/mL).
‡Anti-CMV IgG titre level ≥10 EU/mL.
§Model 1 includes each component separately. Model 2 includes all four components together. All models adjusted for sex, maternal and paternal age at offspring birth, and maternal, paternal and offspring smoking.
¶Coefficient=change in anti-CMV IgG titre level per increase in one unit of component (distribution characteristics in online supplementary 3); RR=change in risk estimate of CMV seropositivity per increase in one unit of component—calculated by Poisson regression with robust error variance.
CMV, cytomegalovirus; RR, relative risk.

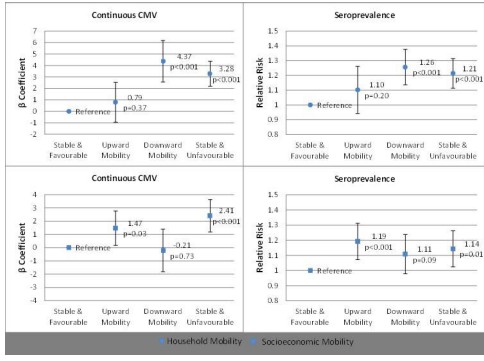

**Figure 1** Predicted mean anti-CMV IgG titre differences and CMV seropositivity risk by household and socioeconomic mobility. Predicted mean differences calculated via β coefficients from linear regression models adjusted for sex, maternal and paternal age at offspring birth, and maternal, paternal and offspring smoking. Relative risk estimation by Poisson regression with robust error variance models adjusted for sex, maternal and paternal age at offspring birth, and maternal, paternal and offspring smoking. CMV, cytomegalovirus.

maintained and associations between socioeconomic components and anti-CMV IgG titre level were attenuated (online supplementary 5). Models looking at associations between individual socioeconomic and household-related variables with anti-CMV IgG titre level found similar results to PCA-based models (online supplementary 6). We also investigated associations between household and socioeconomic component mobility and anti-CMV IgG titre, adjusted for sex, parental ages at offspring birth, and parental and offspring smoking (figure 1). Compared with individuals with stable favourable household components, mean anti-CMV IgG levels were higher in those who maintained unfavourable household components from childhood to adulthood (β=3.23, P<0.001) and those who transitioned from a favourable to unfavourable position (ie, downward mobility) (β=4.32, P<0.001). Those who maintained a stable unfavourable socioeconomic component position had increased anti-CMV titre level compared with those in the stable favourable position (β=2.20, P=0.001) as did those in the upward mobility category (β=1.37, P=0.04). A similar pattern of associations was seen when investigating associations between mobility and CMV seroprevalence (figure 1). To distinguish mobility effects from main effects of childhood and adulthood components, each mobility model was further adjusted—once for the respective (household or socioeconomic) childhood component and separately for the respective adulthood component. Associations between household mobility and anti-CMV titre level as well as with seroprevalence were maintained, while associations between socioeconomic mobility and anti-CMV titre level as well as with seroprevalence were attenuated when the models were adjusted for the socioeconomic childhood component (data not shown). Among CMV seropositive individuals, similar associations were found between household component mobility and anti-CMV

IgG titre and associations between socioeconomic component mobility and anti-CMV IgG titre level were primarily attenuated (online supplementary 7).

## DISCUSSION

This study demonstrated that lower levels of household and socioeconomic components in childhood and adulthood, reflecting lower SEP, were associated with higher anti-CMV IgG titre level at age 32. Additionally, maintaining a stable unfavourable position from childhood to adulthood was associated with higher anti-CMV IgG titre level compared with the stable favourable position. Our study also provides some evidence that social mobility transitions from childhood to adulthood (ie, upward and downward mobility) may be associated with CMV response.

### Life course models

Life course models explaining the impact of SEP on adult health proposed in the literature include the critical period model, emphasising timing of exposure,[48 49] the social mobility model, focusing on transitions across the life course,[48 49] and the accumulation model, emphasising duration of exposures.[48 49] As per the critical period model, our findings that lower childhood and adulthood household and socioeconomic components are associated with higher anti-CMV IgG titre level in early adulthood are in accordance with other studies.[17 27] Our study also adds to the social mobility model, though unlike others,[24 28 35] we found that upward socioeconomic mobility was associated with higher CMV titre and higher risk for CMV seropositivity. Possibly, as others, including JPS analyses, have found,[29] mechanisms similar to those in Gluckman's 'Mismatch hypothesis' are at play—in which early environment induces long-term cardiovascular effects if there is a 'mismatch' (including improvement) between early developmental environment and subsequent environment in later life. Additionally, the attenuation of the associations between upward socioeconomic mobility and CMV with further adjustment for the socioeconomic component at baseline suggests that the socioeconomic environment follows a critical period model as opposed to a social mobility model. Household downward mobility, as in other cohort-based research,[25] was associated with increased CMV IgG titre levels and increased risk for CMV seropositivity, even after adjustment for household components at childhood or adulthood, suggesting that the household environment has a mobility effect above and beyond that of a critical period. Our study is the first to investigate social mobility associations with young adult CMV response.

Research has indicated that sustained exposure to low SEP has been most commonly associated with worse health.[25 28 35–37 48] Our study supports the detrimental impact of accumulation of lower SEP over the life course, as we found increased anti-CMV IgG titre and risk for CMV seropositivity among those in both the household and

socioeconomic stable unfavourable group as compared with stable favourable.

## Multidimensionality of exposure

The associations of social mobility with young adult CMV response differ somewhat in household and socioeconomic components. While household downward mobility was associated with increased anti-CMV IgG titre level, upward socioeconomic mobility seemingly had a harmful impact on immune response at age 32. These findings highlight the multidimensionality of socioeconomic measurement and the varying contribution of different socioeconomic dimensions through the life course.[30] The PCA used in our analyses may provide a more aggregate characterisation of the socioeconomic experience/environment, while demonstrating that various socioeconomic factors may indeed represent different domains. To further highlight this point, household upward or downward mobility comprised only 14.3% of the study population while socioeconomic mobility was seen among 31.1%, possibly indicating that the household component is more stable than the socioeconomic component. Possibly, people tend to maintain a level of religiosity similar to that with which they were raised[50] whereas socioeconomic transitions from parents to offspring vary depending on culture, environment, psychological elements and geographic residence.[51]

## Epigenetic mechanism

Previous studies found that participants originating from low and high SEPs have different patterns of methylation.[52] Additionally, associations have been found between methylation and chronic inflammation and a previous analysis of JPS demonstrated an *HSD11B2*–cardiometabolic risk association.[23] Epigenetic changes in response to environment may play an important role in the development of adult diseases.[53] Further studies that investigate an epigenetic role in the SEP–CMV associations may provide evidence for mechanisms that explain these associations.

## Study strengths and limitations

One of the major strengths of this study is the novel use of CMV antibody, emerging as a potential biomarker for atherosclerosis and cardiometabolic risk, as well as the use of continuous CMV antibody in addition to CMV seropositivity as a study outcome. Few studies have examined CMV antibody titre on a continuous scale, despite potential epidemiological significance of such an outcome. Additionally, very few studies have had the ability to examine the impact of SEP and social mobility on immune response in young adulthood. The combination of high-quality detailed records of perinatal maternal and offspring characteristics and long-term follow-up data at age 32 improved the characterisation of the socioeconomic environment surrounding pregnancy, childhood and adulthood, and allowed analyses to examine changes from childhood to adulthood.

Estimating different components of the socioeconomic environment and trajectory by PCA was another novel approach, enabling us to capture common variance and providing an aggregate view of the socioeconomic experience/environment, rather than limiting analyses to individual SEP proxies.

Our study has several limitations. First, it includes only a sample of offspring from the original 1974–1976 JPS cohort invited to participate in the follow-up study. However, using a stratified sampling approach and oversampling ends of the distribution ensured that offspring with a range of fetal and early-life characteristics were included in our study. Second, examining CMV titre on a continuous scale using qualitative ELISA assay limits clinical interpretation of findings. However, our findings using CMV seropositivity, a clinically relevant diagnosis, support the associations found with CMV antibody titre, and substantiate the use of continuous CMV antibody in future epidemiological studies. Importantly, this specific assay was successfully used in other cohorts, such as the Multi-Ethnic Study of Atherosclerosis.[12] It is important to note that while the immune response to CMV has an extensive impact on the immune system, using CMV antibody titre as a proxy measure of the general cellular immune condition may not be ideal, particularly given the unique cellular response to CMV from the host. More modern approaches, using multiple technological platforms, where the multiple responding cells and cytokines in the blood can be measured,[54] may give a more in-depth understanding of the overall immune response. Additionally, data consist of a single CMV titre measure for each participant. Verification of titre level from additional tests or other sources was not available, and timing/age at initial seroconversion is unknown. However, research has indicated that CMV antibody titre remains relatively stable over time,[55 56] and seroprevalence rates in our population fall in the range of prevalence rates in similar populations.[2 15 16 57] Furthermore, the use of PCA may mask distinctions in relationships between individual SEP variables and CMV. However, additional analyses revealed that associations with the individual SEP variables are in line with what has been previously reported in the literature,[32] and similar to PCA-based results found negative associations between education and anti-CMV IgG titre level, particularly in childhood, and positive associations between religiosity and family size with anti-CMV IgG titre particularly in adulthood. PCA was used to reduce the number of potentially correlated variables in our cohort to a smaller number of principal components that would produce a more coherent picture of the SEP–CMV relationship. While relationships between SEP and CMV have been reported in studies in other populations, the differing associations found in our study between socioeconomic versus household components may be particular to unique elements in Israeli society.

## CONCLUSIONS

Our study extends accumulating evidence of relationships between socioeconomic factors in childhood and adulthood with adult immune response. Additionally, our study provides evidence that accumulating low SEP from childhood through adulthood and social mobility may compromise immune response in young adulthood. Further studies should explore mechanisms underlying the relationships between childhood environments and social mobility with immunity in adulthood. A better understanding of these relationships may lead to new approaches to counteract effects of low-SEP environments through the life course and improve immune response in high-risk young adults.

**Author affiliations**
[1]Braun School of Public Health, Hebrew University-Hadassah Medical Center, Jerusalem, Israel
[2]Cardiovascular Health Research Unit, University of Washington, Seattle, Washington, USA
[3]Department of Epidemiology, University of Washington, Seattle, Washington, USA
[4]Institute for Health and Social Policy, McGill University, Montreal, Canada
[5]Departments of Pathology and Biochemistry, University of Vermont, Burlington, Vermont, USA
[6]Institute for Urban Health, New York Academy of Medicine, New York City, New York, USA

**Contributors** DSS, HH and YF conceived and designed the study. HH designed the data collection tools and developed the protocols. GML designed the analysis plan, conducted the analysis and wrote the manuscript with guidance from YF and HH. DSS and RPT advised on the clinical context of the study and analyses. OM, RCM, DAE and JYH advised on data analysis and interpretation of findings. All authors contributed intellectually to the manuscript, reviewed and edited the manuscript, and approved the final version.

**Funding** This work was financially supported by the National Institutes of Health (grant numbers R01HL088884 and K01HL103174) and the Israeli Science Foundation (grant numbers 1252/07 and 552/12).

**Competing interests** None declared.

**Patient consent** Obtained.

**Ethics approval** This study was approved by the Institutional Review Board of the Hadassah-Hebrew University Medical Center and by the University of Washington Human Subject Review Committee.

**Provenance and peer review** Not commissioned; externally peer reviewed.

**Data sharing statement** Data are available upon request to HH (hagith@ekmd.huji.ac.il).

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
