## [Reviewer comments · BMJ Open]

ARTICLE DETAILS

TITLE (PROVISIONAL)	Associations of social environment, socioeconomic position and social mobility with immune response in young adults: The Jerusalem Perinatal Family Follow-up Study
AUTHORS	Lawrence, Gabriella; Friedlander, Yehiel; Calderon-Margalit, Ronit; Enquobahrie, Daniel; Huang, Jonathan; Tracy, Russell; Manor, Orly; Siscovick, David; Hochner, Hagit

VERSION 1 – REVIEW

REVIEWER	David Strogatz Bassett Research Institute, Bassett Healthcare Network, Cooperstown, NY, USA
REVIEW RETURNED	11-Apr-2017

GENERAL COMMENTS	My primary concern with this manuscript is the PCA-derived household measure. I'm not sure how to interpret a measure combining religiosity with number of siblings or offspring, though I would not consider it any kind of indicator of socioeconomic position, even taking a broad view of SEP (p. 5). I also don't see the inherent directionality of the measure, i.e. how being less religious and/or having fewer children should be considered an improvement or upwardly mobile (p. 6). Last but not least, what is the mechanism that might link the composite measure to anti-CMV IgG titer level? The sibling/offspring measures by themselves might reflect household crowding. There is no discussion of the association with religiosity and as mentioned above, it's not intuitive why these variables should be combined (despite what the PCA produced). PCA is especially valuable when condensing a longer list of related measures into a single composite measure. There seems to be modest gain here when only two indicators of SEP (education and occupation) are reduced to one score, while sacrificing the original, more interpretable metrics (years of education, occupation types presumably represented by the 6 levels of the occupation variable). Analyses of the separate education and occupation variables (perhaps retaining more than 3 levels of occupation) would be clearer, may reveal distinctions concealed by the PCA measure (e.g. stronger influence of education?) and would facilitate comparison with results from the NHANES papers (reference 24 and the seroprevalence paper by Dowd et al., Epidemiol Infect 2009;137:58-65) and other publications that present findings by separate indicators of SEP.
---

REVIEWER	Rebecca Stebbins The Gillings School of Global Public Health The University of North Carolina at Chapel Hill United States of America
REVIEW RETURNED	04-Jul-2017

GENERAL COMMENTS	This paper describes a well-executed analysis investigated the relationship between socioeconomic status and socioeconomic trajectory from childhood to young adulthood and immune response to cytomegalovirus. The authors present an effective analysis of this question using a sample and appropriate analysis techniques to address their research question. However, they should include the following in their discussion of the limitations of the article: 1) a discussion of the limitations of their measurement of /analysis variable for SES, 2) a description of the generalizability of their results, and 3) a mention of the possible bias induced by unmeasured confounding.
--

REVIEWER	Jonathan D. Turner Department of Infection and Immunity, Luxembourg Institute of Health, Esch sur Alzette, Grand Duchy of Luxembourg
REVIEW RETURNED	31-Jul-2017

GENERAL COMMENTS	The manuscript of Lawrence et al. addresses the impact of socioeconomic position (SEP) in childhood and adulthood on cell mediated immune function as measured by CMV antibody titres in a subset of a large birth cohort from Jerusalem. As correctly identified by the authors, the main strength of this study is the large population-based birth cohort together with its rich archive of longitudinal data. This has permitted a very nice characterisation of both the childhood and adult socioeconomic environment, and especially how this has evolved over time. The manuscript does not include the mandatory CONSORT, PROBE or PRISMA checklist, however, this data is available in previous cohort publications. There are however numerous points that need addressing, particularly the data analysis, before the manuscript can be considered for publication.  1. The authors are correct in that few studies use CMV antibody titres as a continuous scale, however, this measure depends on having been previously infected with CMV. The majority of these previous study have found a strong link with to serostatus. In this study, CMV seronegative participants seem to have been included in the analysis of CMV titres. This causes a bias in the results, especially because the authors report an association between CMV serostatus and socioeconomic position. Consequently, the reported association between CMV titres including seronegative participants will most probably only reflect this association. Consequently, these data may not reflect immune function and therefore the results (with the current analysis) do not support the authors conclusion that they found a relationship between SEP and immune response. 2. I strongly encourage the authors to discuss the limitations of using antibody titres to members of the herpesviridae, which are at best a proxy measure of cellular immune condition, and likely not a very good one that should be superseded by more modern approaches.
--

	Minor comments: 1. the abbreviation SEP is not introduced in the abstract.2. Large numbers are not consistently written (1,300 vs 1300 vs 1300).3. The authors need to clearly differentiate in the text between 'CMV serostatus' or 'CMV titers'. It is not always clear to what 'a association with CMV' refers to.
--	--

VERSION 1 – AUTHOR RESPONSE

Reviewer: 1

Reviewer Name@: David Strogatz

Institution and Country: Bassett Research Institute, Bassett Healthcare Network, Cooperstown, NY, USA

Please state any competing interests: none declared

Please leave your comments for the authors below

Comment:

My primary concern with this manuscript is the PCA-derived household measure. I'm not sure how to interpret a measure combining religiosity with number of siblings or offspring, though I would not consider it any kind of indicator of socioeconomic position, even taking a broad view of SEP (p. 5). I also don't see the inherent directionality of the measure, i.e. how being less religious and/or having fewer children should be considered an improvement or upwardly mobile (p. 6). Last but not least, what is the mechanism that might link the composite measure to anti-CMV IgG titer level? The sibling/offspring measures by themselves might reflect household crowding. There is no discussion of the association with religiosity and as mentioned above, it's not intuitive why these variables should be combined (despite what the PCA produced).

PCA is especially valuable when condensing a longer list of related measures into a single composite measure. There seems to be modest gain here when only two indicators of SEP (education and occupation) are reduced to one score, while sacrificing the original, more interpretable metrics (years of education, occupation types presumably represented by the 6 levels of the occupation variable). Analyses of the separate education and occupation variables (perhaps retaining more than 3 levels of occupation) would be clearer, may reveal distinctions concealed by the PCA measure (e.g. stronger influence of education?) and would facilitate comparison with results from the NHANES papers (reference 24 and the seroprevalence paper by Dowd et al., *Epidemiol Infect* 2009;137:58-65) and other publications that present findings by separate indicators of SEP.

Response:

Thank you Dr. Strogatz for your insights and very thoughtful and thorough feedback. We carefully considered each of your points, and decided to add in-depth follow-up analyses with the individual SEP and social environment variables. As the results were actually quite similar to the results using PCA, we added these analyses in the manuscript supplement (supplement 6). Additions were made throughout the manuscript (statistical analyses, page 8, last sentence and top of page 9; results, paragraph 3, page 11, second sentence; discussion, page 16, sentences 1-3) to reflect the additional analyses.

As previously mentioned in the manuscript (page 6) and with additional clarification on page 4, the multi-ethnic society of Israel provides an opportunity to examine SEP alongside culturally relevant social environment characteristics, including religiosity and family-size in addition to well-recognized SEP variables such as occupation and education.

Following the successful use of PCA in creating socioeconomic indices in past and current studies (Vyas and Kumaranayake, 2006 – reference 42 in manuscript; Becher et al., 2016 – reference 43 in manuscript), and given the array of SEP-related variables in the literature, and the known overlap and correlation between many SEP-related variables, we used PCA to reduce the number of potentially correlated variables in our cohort to a smaller number of principle components that would produce a more coherent picture of the SEP-CMV relationship. While it is indeed true that we do not have an expansive list of SEP-related variables, we do look at two different time points, creating an array of potentially related variables. Additionally, as social mobility was of primary interest, our perspective was that reporting associations between multiple SEP variables individually with anti-CMV IgG titer level, and following with multiple social trajectory associations would be extremely difficult to follow. Estimating different components of the socioeconomic environment and trajectory by PCA was a useful approach, enabling us to capture common variance and providing an aggregate view of the socioeconomic experience/environment, rather than limiting analyses to individual SEP proxies. To help clarify and justify our use of PCA, we added relevant references and information on page 7, principle component analysis section, sentence 1.

In addition to capturing the largest possible variance, these components nicely portray two elements of SEP in Israeli society – the socioeconomic element characterized by education and occupation as is often the case in many societies, and the social element, unique to Israeli society, in which families that identify as more religious tend to have more children, with the most religious families often among the lower income levels. The State of the Nation Report, 2014 revealed that of all population sectors in Israel, ultra-orthodox Jews have the highest monthly deficit when looking at the gap between total household income and total expenditures. This may be related to the fact that the total fertility rate averaged 6.9 children per woman in the ultra-orthodox community compared to 3.1 in the general Israeli population (Statistical Report on Ultra-Orthodox Society in Israel, 2016). Therefore, it makes sense that in our cohort, higher component score indicates improvement in SEP – i.e. higher household component score indicates fewer children/decreased religiosity and higher socioeconomic component score indicates higher occupation level/years of education. We added clarification to this effect on page 7, last sentence.

Reviewer: 2

Reviewer Name: Rebecca Stebbins

Institution and Country: The Gillings School of Global Public Health, The University of North Carolina at Chapel Hill

United States of America

Please state any competing interests: None declared.

Please leave your comments for the authors below

This paper describes a well-executed analysis investigated the relationship between socioeconomic status and socioeconomic trajectory from childhood to young adulthood and immune response to cytomegalovirus. The authors present an effective analysis of this question using a sample and appropriate analysis techniques to address their research question. However, they should include the following in their discussion of the limitations of the article:

- 1) a discussion of the limitations of their measurement of /analysis variable for SES,
- 2) a description of the generalizability of their results, and
- 3) a mention of the possible bias induced by unmeasured confounding.

Response: Thank you Ms. Stebbins for taking time to review our manuscript. As per your guidance, we have added to the limitations of the manuscript (page 16).

Reviewer: 3

Reviewer Name: Jonathan D. Turner

Institution and Country: Department of Infection and Immunity, Luxembourg Institute of Health, Esch sur Alzette,

Grand Duchy of Luxembourg

Please state any competing interests: None declared

Please leave your comments for the authors below

The manuscript of Lawrence et al. addresses the impact of socioeconomic position (SEP) in childhood and adulthood on cell mediated immune function as measured by CMV antibody titres in a subset of a large birth cohort from Jerusalem. As correctly identified by the authors, the main strength of this study is the large population-based birth cohort together with its rich archive of longitudinal data. This has permitted a very nice characterisation of both the childhood and adult socioeconomic environment, and especially how this has evolved over time. The manuscript does not include the mandatory CONSORT, PROBE or PRISMA checklist, however, this data is available in previous cohort publications. There are however numerous points that need addressing, particularly the data analysis, before the manuscript can be considered for publication.

1. The authors are correct in that few studies use CMV antibody titres as a continuous scale, however, this measure depends on having been previously infected with CMV. The majority of these previous study have found a strong link with to serostatus. In this study, CMV seronegative participants seem to have been included in the analysis of CMV titres. This causes a bias in the results, especially because the authors report an association between CMV serostatus and socioeconomic position. Consequently, the reported association between CMV titres including seronegative participants will most probably only reflect this association. Consequently, these data may not reflect immune function and therefore the results (with the current analysis) do not support the authors conclusion that they found a relationship between SEP and immune response.

Response: Thank you Dr. Turner for your insights. To assess whether including seronegative participants bias the results by merely reflecting the association between serostatus and SEP we re-ran all analyses among seropositive individuals only. Among the seropositive individuals, we found very similar results in associations between household components and mobility with anti-CMV IgG titer and while results between socioeconomic components and mobility with anti-CMV IgG titer were primarily attenuated, the direction and pattern in coefficients were maintained. We have added these results to the manuscript (see changes in methods, page 9 – end of first paragraph, and results, page 10, end of first paragraph, page 11, top of page and bottom of page as well as supplements 4, 5 and 7.)

2. I strongly encourage the authors to discuss the limitations of using antibody titres to members of the herpesviridae, which are at best a proxy measure of cellular immune condition, and likely not a very good one that should be superseded by more modern approaches.

Response: As per your guidance, we have added to the discussion section (page 15, mid-paragraph).

Minor comments:

1. the abbreviation SEP is not introduced in the abstract.
2. Large numbers are not consistently written (1,300 vs 1300 vs 1 300).
3. The authors need to clearly differentiate in the text between 'CMV serostatus' or 'CMV titers'. It is not always clear to what 'a association with CMV' refers to.

Response: Noted and revised throughout manuscript

VERSION 2 – REVIEW

REVIEWER	David Strogatz Bassett Research Institute, Bassett Healthcare Network, Cooperstown, NY, USA
REVIEW RETURNED	22-Oct-2017

GENERAL COMMENTS	I appreciate the authors' responses to the previous review of the manuscript. In particular, the additional information on the household measure and the rationale for PCA is very helpful. I have a few remaining comments, with references to the page and line numbers in the version of the manuscript without track changes. 1. page 5, lines 8-11: The statement that religious observance impacts health "irrespective of socioeconomic position" seems awkward when on page 8, lines 26-31 the household component is described as one of "two elements of SEP in Israeli society." An alternative wording for the end of the sentence on page 5 might be "... religious observance, a central characteristic of the Israeli social environment, contributes independently to social position and adult health [31]."2. page 8, lines 23-24: Another alternative wording to suggest, since occupation and education are not environmental measures per se - "... and the socioeconomic environment represented by parental and offspring education and occupation."3. A lingering methodological concern with implications for the interpretation and discussion of results, now that I have a better understanding of the household component. Is there a formal test that distinguishes mobility effects from main effects of childhood and adult status? Stated more directly, do these results show a main effect of low childhood SE environment (independent of adult SE environment) and a main effect of low adult HH environment (independent of childhood HH environment)? So that the results speak to critical period and accumulation models of life course effects, but perhaps not additional evidence related to mobility?4. Minor edits: "principle" should be "principal" on page 8, line 13 and page 20, line 11. On page 8, line 41 insert the word "are" between "PCA" and "provided." On page 10, line 33, 7.2% should be 7.1%. Also, are the references 31 and 47 in the reference list complete?
--

REVIEWER	Rebecca Stebbins UNC - Chapel Hill USA
REVIEW RETURNED	28-Oct-2017

GENERAL COMMENTS	This paper contributes to an important and developing field of literature looking at persistent pathogens and how they may be a mediator on the causal pathway from life course socioeconomic to health outcomes, including CVD and atherosclerosis. The authors do a fine job of assessing their research question, with appropriate statistical methods. They do a particularly good job considering the multi-dimensionality of socioeconomic position by creating two separate constructs based on several factors, investigating the change in those over time, and discussing the implications of their results. The authors of adequately responded to reviewer comments from the previous review. In particular, their additions to the limitations section of their discussion were necessary and how provide a more comprehensive assessment of their work's limitations. The additions of discussions on the limitations of CMV antibody titer as a proxy measure for cellular immunity and the use of PCA were welcome.
---

REVIEWER	Jonathan D. Turner Luxembourg Institute of Health Department of Infection and Immunity, Esch-sur-Alzette, Grand Duchy of Luxembourg
REVIEW RETURNED	04-Oct-2017

GENERAL COMMENTS	The authors have adequately addressed my concerns.
--

VERSION 2 – AUTHOR RESPONSE

1. page 5, lines 8-11: The statement that religious observance impacts health "irrespective of socioeconomic position" seems awkward when on page 8, lines 26-31 the household component is described as one of "two elements of SEP in Israeli society." An alternative wording for the end of the sentence on page 5 might be "... religious observance, a central characteristic of the Israeli social environment, contributes independently to social position and adult health [31]."

Response: Revised as suggested (page 4)

2. page 8, lines 23-24: Another alternative wording to suggest, since occupation and education are not environmental measures per se - "... and the socioeconomic environment represented by parental and offspring education and occupation."

Response: Revised as suggested (page 7)

3. A lingering methodological concern with implications for the interpretation and discussion of results, now that I have a better understanding of the household component. Is there a formal test that distinguishes mobility effects from main effects of childhood and adult status?

Stated more directly, do these results show a main effect of low childhood SE environment (independent of adult SE environment) and a main effect of low adult HH environment (independent of childhood HH environment)? So that the results speak to critical period and accumulation models of life course effects, but perhaps not additional evidence related to mobility?

Response: Thank you for this excellent and insightful comment. We have run additional models to formally distinguish mobility effects from main effects of childhood and adulthood (see results on bottom of page 15), and now include some additions to the interpretation of the findings (discussion page 17, first paragraph).

4. Minor edits: "principle" should be "principal" on page 8, line 13 and page 20, line 11. On page 8, line 41 insert the word "are" between "PCA" and "provided." On page 10, line 33, 7.2% should be 7.1%. Also, are the references 31 and 47 in the reference list complete?

Response: All revised as suggested

VERSION 3 – REVIEW

REVIEWER	David Strogatz Bassett Research Institute, Bassett Healthcare Network, Cooperstown, NY, USA
REVIEW RETURNED	16-Nov-2017
GENERAL COMMENTS	Thanks to the authors for their consideration of our comments and suggestions.